# A Study of Optimal Specifications for Light Shelves with Photovoltaic Modules to Improve Indoor Comfort and Save Building Energy

**DOI:** 10.3390/ijerph18052574

**Published:** 2021-03-04

**Authors:** Heangwoo Lee, Xiaolong Zhao, Janghoo Seo

**Affiliations:** 1Major in Spatial Design, College of Design, Sangmyung University, Cheonan-si 31066, Chungcheongnam-do, Korea; 2hw@smu.ac.kr; 2Department of Design, College of Design, Sangmyung University, Cheonan-si 31066, Chungcheongnam-do, Korea; whgyfyd@naver.com; 3School of Architecture, Kookmin University, 77, Jeongneung-ro, Seongbuk-gu, Seoul 02707, Korea

**Keywords:** light shelf, PV module, thermal environment, visual comfort, performance evaluation

## Abstract

Recent studies on light shelves found that building energy efficiency could be maximized by applying photovoltaic (PV) modules to light shelf reflectors. Although PV modules generate a substantial amount of heat and change the consumption of indoor heating and cooling energy, performance evaluations carried out thus far have not considered these factors. This study validated the effectiveness of PV module light shelves and determined optimal specifications while considering heating and cooling energy savings. A full-scale testbed was built to evaluate performance according to light shelf variables. The uniformity ratio was found to improve according to the light shelf angle value and decreased as the PV module installation area increased. It was determined that PV modules should be considered in the design of light shelves as their daylighting and concentration efficiency change according to their angles. PV modules installed on light shelves were also found to change the indoor cooling and heating environment; the degree of such change increased as the area of the PV module increased. Lastly, light shelf specifications for reducing building energy, including heating and cooling energy, were not found to apply to PV modules since PV modules on light shelf reflectors increase building energy consumption.

## 1. Introduction

In addition to the depletion of major energy sources, the rise in energy consumption has become a threat to our society, causing a range of environmental problems [1,2]. As such, the demand for research and technology development to save energy is growing throughout our society. According to the “2019 Global Status Report for Buildings and Construction” by the Global Alliance for Building and Construction, the amount of energy used to create a comfortable indoor environment in the building sector accounted for 30% of all energy consumption [3]. In addition, according to the United States (US) Department of Energy’s “2015 Renewable Energy Data Book”, buildings accounted for 38.9% of all energy consumption in the US and, as a result, research on reducing building energy consumption is becoming more and more important [4]. In detail, the energy used for space heating, space cooling, and lighting accounts for 20.8%, 10.0%, and 11.3% of building energy consumption, respectively. Therefore, these factors must be considered in research and technology development to save building energy. Among these technologies for saving building energy, a light shelf is a typical natural daylighting system that works as building envelope element technology installed in windows to introduce external natural light deep into a room through the use of reflection. Various studies have confirmed the effectiveness of this technology [5,6]. Previous studies related to light shelves have expanded from performance evaluations with installation of light shelves to research integrating advanced information technology (IT), such as user recognition and location recognition technologies [7,8]. A photovoltaic (PV) module is a type of renewable energy technology used to produce electric energy from natural light. These modules have recently been applied in various areas, including building envelopes [9,10]. However, PV modules generate a substantial amount of heat while generating power, which reduces the generating efficiency of the modules [11].

A recent study on light shelves found that efficient building energy savings could be maximized by integrating a PV module with the light shelf reflector to perform simultaneous daylighting and light concentration [12]. However, this study only considered the concentration of light by the light shelf and PV module and did not consider the substantial amount of heat generated by the PV module while concentrating light. The heat generated by PV modules significantly affects the thermal environment of the indoor space because light shelves are installed in windows with relatively low insulation performance. This generated heat changes the consumption of heating and cooling energy, as well as the PV module specifications applied to light shelves.

Therefore, the purpose of the current study was to evaluate building energy performance by considering the amount of heat generated by PV module light shelves (or light shelves incorporating PV modules), during daylighting, generation, and concentration processes. These results can validate PV module effectiveness in saving building energy and help derive optimal specifications. Additionally, these results will generate fundamental data that can be used in the design of future light shelves and differentiates our study from prior research.

### 1.1. The Concept and Research Trends of Light Shelves

A light shelf is a natural daylighting system installed inside or outside a window to introduce natural light into a room. As shown in Figure 1, light shelves introduce natural light deep into the room by reflecting light from the reflector of the light shelf and the ceiling of the indoor space. This process improves the indoor lighting environment and saves lighting energy [13,14]. In addition, light shelves can improve the balance of indoor lighting by blocking excessive natural light from entering the window during the daytime [15]. Light shelves can also induce thermal changes in an indoor space by blocking natural light flowing through the window. As shown in Figure 2, the installation height and the angle and reflectance of the light shelf reflector are the primary indicators that determine daylighting performance of light shelves. Light shelves can be divided into external and internal types depending on whether they are installed outside or inside the window.

As shown in Table 1, previous studies related to light shelves [5,6,7,8,12,16,17,18,19,20,21,22,23] controlled the variables associated with light shelves or changed their shape to improve daylighting performance. Recent studies integrated various moving or operational technologies, including IT, and applied other elements to light shelves, such as awnings and PV modules. A study that combined dual PV modules with a light shelf [17] indicated improved building energy reduction efficiency by attaching the module on part of, or in front of, the light shelf reflector. They found that installing PV modules on part of the light shelf, instead of installing the modules in front of the light shelf, was more advantageous in terms of saving building energy [12]. This is because installing PV modules in front of the light shelf only generates energy by concentrating light; hence, it is better to allow both daylighting and concentration when integrating PV modules with light shelves. However, most of the previous studies that integrated PV modules with light shelves focused only on the energy produced by the concentration of the PV modules and the indoor lighting energy savings; the heat generated by the PV module while concentrating light has not been considered. Therefore, this study pursues more effective performance evaluation by reflecting the heat generated by PV module light shelves during daylighting and the concentration process.

### 1.2. PV Module Concept and Power Generation Principle

The PV module is an essential component of all PV systems and consists of an assembly of PV or solar cells that directly converts sunlight into direct current (DC) electricity. In other words, a PV module is an assembly of PV cells connected in series or parallel to deliver the voltage and current required for a specific system. As shown in Figure 3, a PV module consists of a ribbon for connecting PV cells, a frame for affixing the PV cells, a terminal box to extract the generated electricity, and a power cable. A PV cell, the basic unit of a PV module, has a p–n junction, as shown in Figure 4, and the principle by which it generates electrical energy is as follows. When external natural light strikes a PV cell, electrons and holes are generated inside the cell due to the energy of the natural light photons. These electrons and holes generate energy by moving to and loading the n-type and p-type semiconductor via an electric field generated at the p–n junction [12]. PV cells have been in the spotlight as an energy source that can replace fossil fuels because they can continuously produce electricity. However, PV cells become hot during the light concentration process and studies have found that the temperature at the rear part of PV cells exceeds 70 °C when concentrating light to produce electricity [24,25]. The high temperature generated during this process increases the temperature of the indoor space, and research has found that PV modules applied to buildings can increase the indoor temperature by 4 °C, depending on the conditions. Therefore, this temperature increase should be considered when applying PV modules to buildings [11]. In particular, the higher the concentration of light, the higher the temperature of the PV cell, and this high-temperature state reduces the power generation efficiency. As a result, PV modules can generate more power during mid-season compared to summer [26].

### 1.3. Consideration of Optimal Indoor Illumination and Temperature Standards

Optimal indoor illumination and temperature standards were considered in order to evaluate the performance of PV module light shelves. As shown in Table 2 and Table 3, this study considered illumination standards in the US, Japan, Korea, Europe, and China to derive illumination standards for creating a pleasant light environment in indoor spaces. [27,28,29,30,31]. On the basis of these findings, 500 lx was set as the standard for lighting control between performance evaluations by reflecting the intersection (~500–600 lx) for general visual work in the illumination standards in the US, Japan, and Korea, and the illumination required for writing and graphic design work in office spaces in the illumination standards of Europe and China. As shown in Table 4, optimal indoor temperature standards in the US and Korea [32,33] were also considered to determine optimal indoor temperature. According to these results, the optimal indoor temperature for performance evaluation was set to 26 °C and 20 °C for summer and winter, respectively.

## 2. Materials and Methods

### 2.1. PV Module Light Shelf Variable Settings for Performance Evaluation

As shown in Table 5 and Figure 5, to evaluate the performance of saving lighting, heating, and cooling energy by considering daylighting, concentration, and heat generated by a PV module light shelf, the ratio of PV modules installed on the light shelf reflector was controlled to be 0% (Case 2), 33.4% (Case 3), 66.8% (Case 4), and 100% (Case 5). Case 2 refers to a conventional light shelf and Case 5 only generates power by concentrating light, as PV modules in this case completely cover the light shelf reflector. In Cases 3 and 4, the light reflector and PV modules work together to simultaneously perform daylighting and light concentration. The PV modules were installed at the same angle as the light shelf reflector and, as a result, the angle at which natural light enters the PV module changed according to the angle of the light shelf. Therefore, in Cases 2 and 3, the reflector was installed near the moving axis for controlling the light shelf angle to perform daylighting, and PV modules were installed in the remaining area to concentrate light, as shown in Figure 5. This study also validated the effectiveness of light shelves by including Case 1, which had no light shelf. Table 6 shows the specifications of the PV cells in the PV modules in this study, which had a Grade-A efficiency of 18.2%. In terms of the light shelf variables for performance evaluation, the width was set to 0.52 m in consideration of the size of the PV cell, as well as the frame and ribbon constituting the PV module, as shown in Table 7. In consideration of the installation process in actual buildings, the rear part of the PV module did not have a separate cooling system to address the high temperature and heat generated while concentrating light. The height of the light shelf was set to 1.8 m, considering the eye level of people. The light shelf angle was set in 10° increments, from −70° to 30°, by considering the altitude of natural light incident on the PV module and findings from previous studies [12,13].

### 2.2. Performance Evaluation Environment Settings

For this study, a full-scale testbed was built to conduct performance evaluations that validated the effectiveness of PV module light shelves and derived optimal specifications. As shown in Table 7 and Figure 6, the size of the testbed was 4.9 m (W) × 6.6 m (D) × 2.5 m (H), and 100 mm thick insulation panels were used as walls. The size of the window, wherein the light shelf that incorporated a PV module was installed, was 1.9 m (W) × 1.7 m (H) and made of double-glazed glass with 80% transmissivity. Lights and air conditioners were installed inside the testbed to maintain the optimal indoor illumination and temperature. Four light-emitting diode (LED)-type lights, capable of eight-level dimming control, were installed on the ceiling inside the testbed according to the Illuminating Engineering Society (IES) four-point method [27]. The lights applied in this study were prototypes; thus, they did not have a separate model name, but Figure 7 shows their light distribution curve and conical illuminance characteristics. As shown in Figure 8, illuminance and temperature sensors were also installed to collect environmental information of the space inside the testbed. The locations of the illuminance sensors were adjusted on the basis of a study that showed that illuminance measured 4.4 m away from the window was similar to average indoor illuminance [34]. As a result, eight illuminance sensors were installed at 1.1 m intervals. The illuminance sensors were also placed 0.8 m from the floor, considering the height of the work surface. The temperature sensor was installed in the middle of the indoor space.

An artificial climate chamber was built outside of the window where the light shelf was installed. The chamber was equipped with an artificial solar irradiation device to simulate the intensity and solar altitude. It was also possible to adjust the temperature in the chamber. This Grade-A apparatus ensured measurement uniformity, in accordance with ASTM E927-85 [26], to obtain valid results between performance evaluations. However, due to the mechanical characteristics of this device, only the performance when the sun was facing south was evaluated, which is a limitation of this study. Table 8 shows the environment of the chamber for summer, mid-season, and winter, according to a related study in Seoul, Korea [35]. However, the solar irradiation presented in Table 8 is the result of controlling the artificial solar irradiation device and the external temperature during mid-season, which was based on the average temperature in September for the past 30 years according to the Korea Meteorological Administration [36].

### 2.3. Performance Evaluation Method

The performance of the PV module light shelf, in terms of improving indoor comfort and saving building energy, was evaluated as described below.

First, the distribution of indoor illuminance according to the light shelf angle and PV attachment area was investigated to determine the uniformity ratio. Uniformity ratio refers to the degree of uniformity of the indoor illuminance. Unbalanced illuminance increases the unpleasant environment and, thus, should be incorporated into the planning of the indoor space. Furthermore, future studies should evaluate the light shelves considering other variables that affect the performance of the light shelves such as adjusted angles and number of PV modules [29]. The uniformity ratio of indoor illuminance was calculated as the average illuminance for minimum illuminance. This study used AutoCAD to visualize the process of external natural light entering a room through the light shelf to use as data to analyze the performance evaluation results.

Second, the amount of lighting energy used to maintain optimal indoor illuminance in each of the cases was determined according to the method below. After monitoring the values of the illuminance sensors in the indoor space, lighting dimming control was performed only when the minimum value of the sensors was less than 500 lx. Dimming control was performed by raising the dimming level of the light next to the illuminance sensor sequentially from the minimum value level of one to eight, until all of the illuminance sensors reached 500 lx. However, if the minimum value of the illuminance sensor did not reach 500 lx, even after adjusting this light to level eight, dimming control was performed on the next closest light until the illuminance sensors reached 500 lx. The amount of lighting energy used according to the lighting dimming control was determined and performance evaluation was conducted for one day during summer, mid-season, and winter.

Third, the amount of power generated by the PV module was determined. As shown in Figure 9, the amount of power generated by the PV module was calculated by multiplying the voltage and the current of the solar power generated by the PV module. Table 9 shows the specifications of the equipment used to measure the voltage and current of the PV module.

Fourth, the amount of cooling and heating energy used to maintain the optimal indoor temperature for each case was determined, to analyze the effect of the heat generated by the PV module while concentrating light on the indoor thermal environment, using the method below. The air conditioner was controlled in conjunction with the illuminance sensor installed in the center of the indoor space, instead of using the sensor embedded in the conditioner. In cooperation with Samsung SNS (Korea), a system was developed to control the air conditioning and heating equipment to maintain ambient temperatures of 26 °C and 20 °C in summer and winter, respectively. For example, the air conditioner cools down the indoor space if the temperature sensor in the center of the indoor space reaches 28 °C during summer. During this process, when the indoor temperature reaches 26 °C, the air conditioner keeps running with minimum power but does not actually cool down the indoor space. The performance was evaluated by repeating this process and monitoring the power consumption needed to maintain the optimal indoor temperature. However, in mid-season, performance evaluation was conducted without using the air conditioning and heating equipment.

## 3. Results and Discussion

### 3.1. Performance Evaluation Results

The purpose of this study was to validate and derive the optimal specifications of light shelves that incorporate a PV module by analyzing the energy consumption of the lighting, cooling, and heating equipment. The findings are described below.

The indoor illumination distribution and uniformity ratio were examined on the basis of the variables in Table 10. During summer, the uniformity ratios of Cases 2, 3, 4, and 5, which had light shelves, improved by an average of 34% compared to Case 1 with no light shelf, as shown in Table 10. In addition, increasing the light shelf angle during summer was effective in improving the uniformity ratio as the amount of natural light entering the room was increased, as shown in Figure 10 and Figure 11. However, Cases 3, 4, and 5, which had PV modules, were not suitable for improving daylighting performance and uniformity ratio because they had smaller reflector areas than Case 2, which did not have a PV module. In mid-season, increasing the light shelf angle improved the uniformity ratio, as it did in summer. However, as shown in Figure 12, adjusting the light shelf angle to 30° during mid-season was unsuitable because external natural light only flowed into the room directly through the light shelf reflector. The uniformity ratio was reduced and there was an uncomfortable glare. Depending on the angle, during mid-season, the light shelf was also unsuitable for improving the uniformity ratio compared to not having a light shelf. In particular, installing a PV module on the light shelf reflector could reduce both the daylighting performance and the uniformity ratio. In winter, external natural light enters deep into the room through the window due to the low solar altitude compared to summer and mid-season; therefore, installing a light shelf tends to reduce the uniformity ratio by blocking the natural light entering the room. In particular, Cases 3, 4, and 5, which had light shelves, were unsuitable in terms of improving the uniformity ratio compared to Case 1. Adjusting the light shelf angle to 20° during winter was also unsuitable, similar to adjusting the angle to 30° during mid-season, because external natural light directly flowed into the room only through the light shelf reflector, reducing the uniformity ratio and causing discomfort due to glare. When the light shelf angle was adjusted to 30° during winter, as shown in Figure 12, natural light entered the room through the rear part of the light shelf reflector instead of reflecting off the reflector, resulting in a shading effect and a reduced uniformity ratio.

The amount of lighting energy used to maintain optimal indoor illuminance and the energy consumption generated through the PV module are shown in Table 11 and Figure 13, Figure 14 and Figure 15. In summer, increasing the light shelf angle tends to reduce consumption of lighting energy by increasing the amount of natural light entering the room, as shown in Figure 13. However, increasing the PV module installation area reduces the amount of natural light entering the room, thereby increasing the consumption of lighting energy. In addition, the PV module power generation showed the best performance at an angle of −10° during summer. According to these results, the optimal specifications, considering the lighting energy savings and the amount of power generated by the PV module during summer was Case 4 at an angle of −10°, where daylighting and concentration were performed at the same time, as shown in Table 12. In mid-season, increasing the light shelf angle reduced the consumption of building energy, as in summer, and the PV module power generation efficiency showed the best performance at an angle of −40°, as shown in Figure 14. The PV module power generation efficiency was higher due to the relatively low-temperature conditions compared to summer. As a result, Case 5, where the PV module was installed in front of the light shelf reflector, was most suitable for saving building energy, as shown in Table 13. Table 13, which shows the optimal specification for each case during mid-season, does not show the results for when the light shelf angle was 30° because this condition produced an uncomfortable glare. As shown in Figure 15 and Table 14, installing a light shelf during winter was unsuitable for saving energy compared to not installing a light shelf. More natural light enters the room during winter because of the lower solar altitude compared to summer and mid-season, but installing a light shelf blocks this natural light.

Performance of the PV module light shelves was evaluated by considering the lighting energy consumption, as well as the amount of power generated by the PV module, and by analyzing the consumption of cooling and heating energy related to the heat generated by the shading of the light shelf and the concentration of the PV module. In summer, as shown in Figure 16 and Table 15, Case 2 (light shelf without a PV module) reduced the cooling energy by an average of 9.2% compared to Case 1 (no light shelf). The consumption of cooling energy increased as the light shelf angle increased in Case 2, which was caused by a decrease in the shading area and an increase in the amount of natural light entering the room as the light shelf angle increased. In Cases 3, 4, and 5, the amount of cooling energy used to maintain the optimal indoor temperature increased as the PV module area increased during summer. In particular, the consumption of cooling energy was high at −10°, when the PV module generated a large amount of power, and this proved that the heat generated by the PV module while concentrating light affected the indoor thermal environment. According to these results, Case 1 (light shelf with no PV module) was most suitable for saving building energy during summer, which was different from the results of a previous study [12] that only considered daylighting and concentration performance of the PV module-incorporated light shelf. In winter, the heat generated by the PV module was relatively low compared to summer, but this also affected the heating energy of the indoor space. In addition, Cases 2, 3, 4, and 5 were unsuitable for saving building energy because they increased the heating energy by 14.6%, 14.9%, 15.1%, and 15.4%, respectively, compared to Case 1 (no light shelf).

### 3.2. Discussion

This study evaluated the performance of light shelves that incorporate a PV module by considering the consumption of lighting, cooling, and heating energy and the power generated by the PV module to create a pleasant indoor environment. A discussion of the findings is provided below.

First, the area of the light shelf reflector decreased in the light shelf that incorporated a PV module as the area of the PV module increased, which reduced the daylighting performance of the light shelf and increased the consumption of lighting energy. At the same time, increasing the area of the PV module installed in the light shelf reduced the amount of natural light entering the room through reflection, thereby reducing the uniformity ratio. Therefore, the application of a PV module and its application area are significant factors in determining the performance of PV module light shelves. Second, Case 2 (light shelf without PV module) was most suitable for saving building energy, as shown in Table 16. These results differed from the findings of a previous study on PV module light shelves [12], which reported that it is desirable to install PV on a portion of the light shelf reflector to save building energy. This is because the previous study did not reflect the effect of the heat generated by the PV module while concentrating light on the indoor heating and cooling environment. Third, in mid-season, the PV module generated a high amount of power due to the low temperature of the outside air compared to summer, which was effective in terms of saving building energy. However, in winter, not installing a light shelf was more advantageous for saving building energy than installing a light shelf. According to these results, PV module light shelves should be detachable to increase efficiency; however, this feature is difficult to implement in external-type light shelves. Therefore, as shown in Figure 17, the authors of this study propose a light shelf with a detachable moving PV module installed underneath the reflector to improve building energy efficiency. This type of light shelf allows the PV module to be pulled out like a drawer to generate power whenever necessary. However, this light shelf is proposed on the basis of the results and contents of this study; therefore, additional research should be performed to evaluate its performance.

## 4. Conclusions

The purpose of this study was to evaluate the performance of light shelves that incorporate a PV module in improving indoor comfort and saving building energy, to demonstrate their effectiveness, and to derive optimal specifications. The main findings are presented below.

First, in summer and mid-season, installing a light shelf improved the indoor light environment, and increasing the light shelf angle increased the amount of natural light entering the room, improving the uniformity ratio. However, increasing the area of the PV module installed on the light shelf tended to reduce both the daylighting performance and the uniformity ratio. Furthermore, adjusting the light shelf angle to 30° during mid-season was unsuitable because external natural light directly flowed into the room only through the light shelf reflector, thus reducing the uniformity ratio and causing an uncomfortable glare. Installing a light shelf during winter tended to reduce the uniformity ratio due to the relatively low solar altitude compared to summer and mid-season. In addition, adjusting the light shelf angle to 20° during winter was unsuitable due to the uncomfortable glare and reduced uniformity ratio. Adjusting the light shelf angle to 30° was not suitable for improving the uniformity ratio and saving lighting energy because natural light entered the room via the rear part of the light shelf reflector instead of reflecting off the reflector. These results should be considered in the design of light shelves.

Second, this study validated the effectiveness of PV module light shelves by determining the consumption of lighting energy and the amount of power generated by the PV module. The results showed that increasing the light shelf angle during summer and mid-season saved lighting energy by increasing the amount of natural light entering the room. The PV module power generation showed the best performance at angles of −10° and −40° during summer and mid-season, respectively. However, the PV module power generation efficiency was higher in mid-season than in summer due to the relatively low temperature conditions. Therefore, the PV module light shelves showed higher energy-saving efficiency during mid-season. In winter, installing a light shelf was unsuitable for saving energy compared to not installing a light shelf.

Third, the current study analyzed changes in the consumption of heating and cooling energy according to the PV module light shelf. In summer, installing a light shelf reduced the cooling energy required by blocking part of the natural light flowing into the room, but increasing the light shelf angle tended to increase the cooling energy. The heat generated by the concentration of the PV module installed on the light shelf affected the consumption of indoor heating and cooling energy. In addition, increasing the PV module area tended to increase the consumption of heating and cooling energy. Installing a light shelf during winter was unsuitable because it increased the consumption of heating energy.

Fourth, optimal specifications were derived by considering lighting energy, consumption of heating and cooling energy, and power generated by the PV module. The results showed that the conventional light shelf without a PV module was effective in saving building energy, which differs from findings of a previous study, which reported that installing a PV module on part of the light shelf reflector to perform daylighting and concentrate light at the same time was effective in saving building energy. The effectiveness of the light shelf performance evaluation performed in the current study was validated by this result.

This study built on the findings of prior research by studying light shelves that incorporated a PV module to create indoor comfort and save building energy. The proposed light shelf, incorporating a PV module for saving building energy, is also valid as it coincides with the purpose of this study. However, this study did not evaluate the performance of the proposed light shelf that incorporated a PV module; therefore, additional research should be pursued to validate its performance. Performance evaluations should also be conducted in more diverse environments as the current study only evaluated the performance of light shelves incorporating a PV module in an artificial environment. Furthermore, future studies should evaluate the light shelves considering other technical variables that affect the performance of the light shelves such as adjusted angles and number of PV modules, as well as its cost efficiency in practice.

## Figures and Tables

**Figure 1 ijerph-18-02574-f001:**
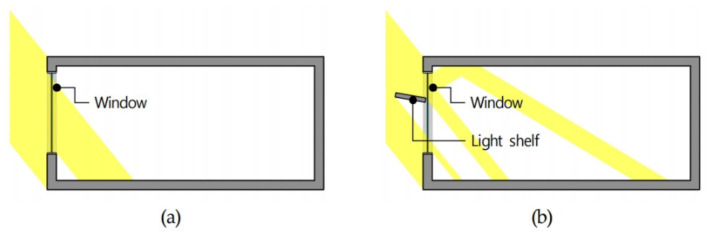
Inflow of light depending on light shelf installation: (**a**) no light shelf and (**b**) light shelf.

**Figure 2 ijerph-18-02574-f002:**
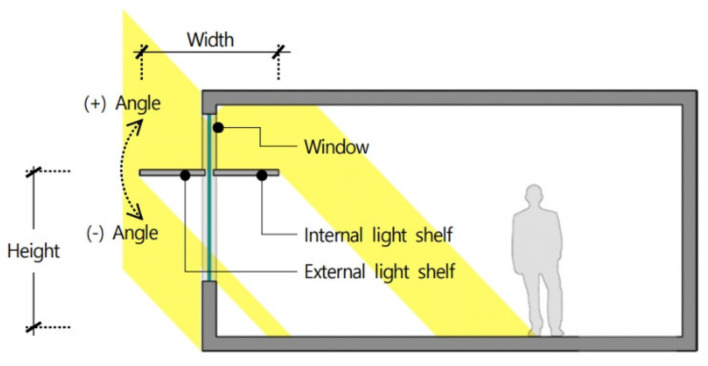
Variables associated with light shelves.

**Figure 3 ijerph-18-02574-f003:**
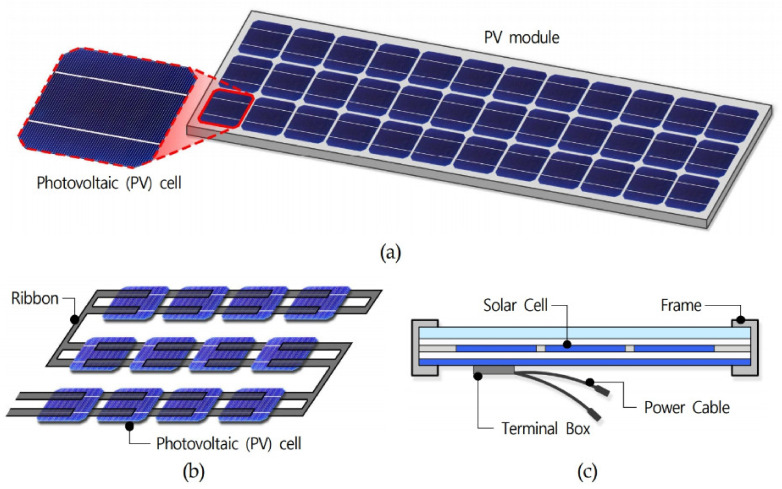
Schematic illustrating variables associated with PV components in light shelves: (**a**) composition of PV modules; (**b**) structure of PV modules; (**c**) section of PV modules.

**Figure 4 ijerph-18-02574-f004:**
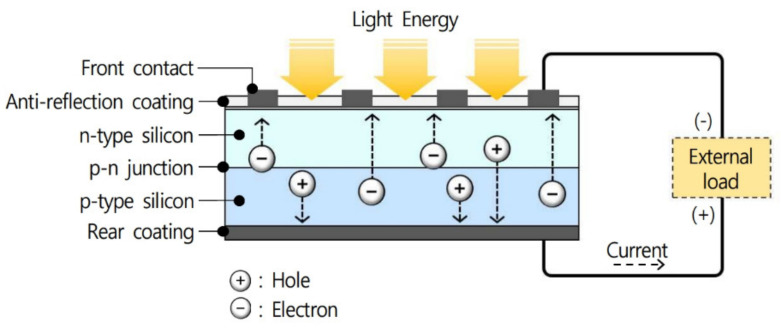
Schematic illustrating electricity generation in PV cells.

**Figure 5 ijerph-18-02574-f005:**
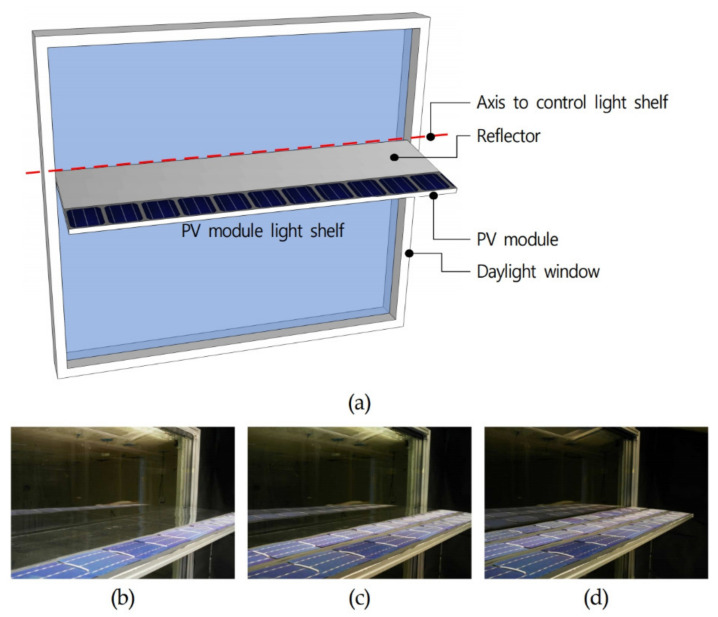
PV module attachment location and case settings for performance evaluation: (**a**) schematic of PV module attachment location; pictures of (**b**) Case 2 (light shelf angle 0°), (**c**) Case 3 (light shelf angle 0°), and (**d**) Case 4 (light shelf angle 0°).

**Figure 6 ijerph-18-02574-f006:**
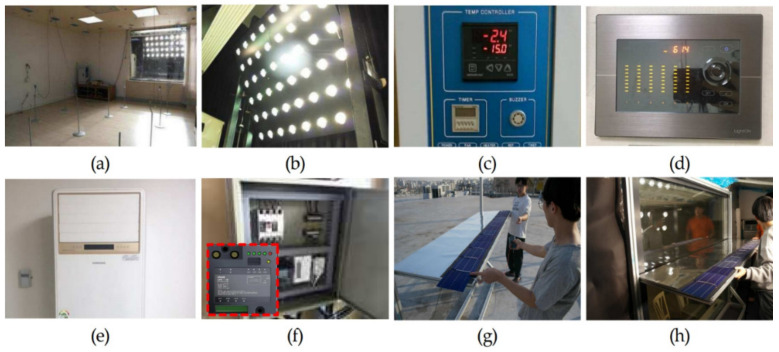
Photographs of testbed and light shelf installation: (**a**) view of testbed; (**b**) artificial solar irradiation apparatus; (c) chamber temperature controller; (**d**) light dimming controller; (**e**) air conditioner; (**f**) energy monitoring system; (**g**) fabricating the light shelf that incorporates a PV module; (**h**) installing the testbed for the light shelf that incorporates a PV module.

**Figure 7 ijerph-18-02574-f007:**
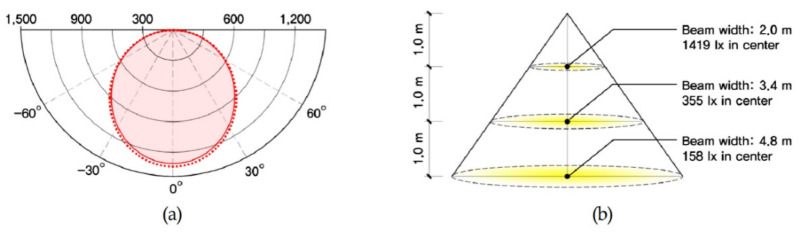
Light distribution curve and conical illuminance of lighting: (**a**) conical illuminance; (**b**) light distribution.

**Figure 8 ijerph-18-02574-f008:**
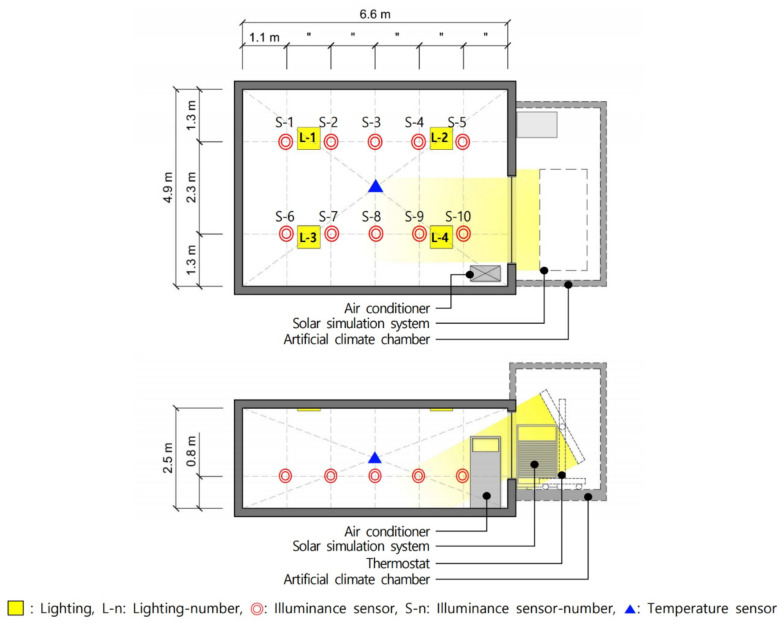
Test bed schematics: plane, section, and sensor locations.

**Figure 9 ijerph-18-02574-f009:**
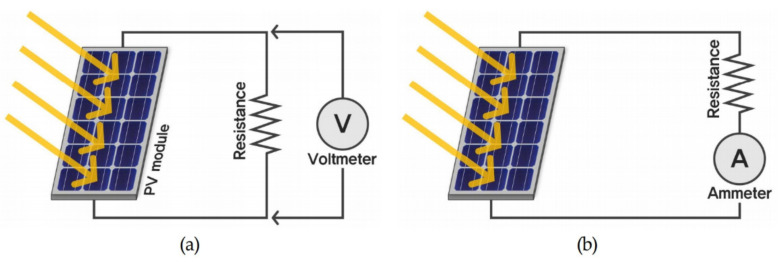
Schematic illustrations of measuring the (**a**) maximum voltage and (**b**) maximum current of the PV module.

**Figure 10 ijerph-18-02574-f010:**
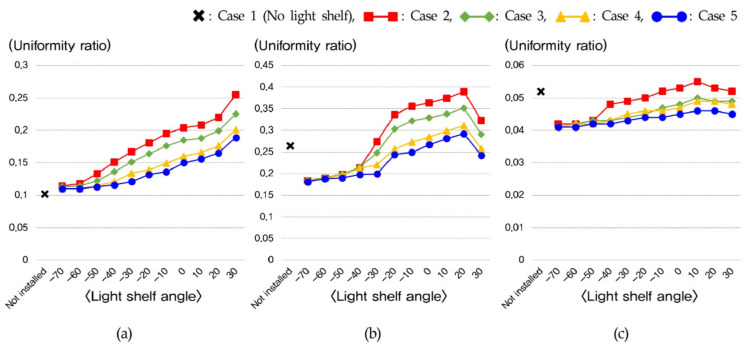
Uniformity ratio according to light shelf angle in (**a**) summer, (**b**) mid-season, and (**c**) winter.

**Figure 11 ijerph-18-02574-f011:**
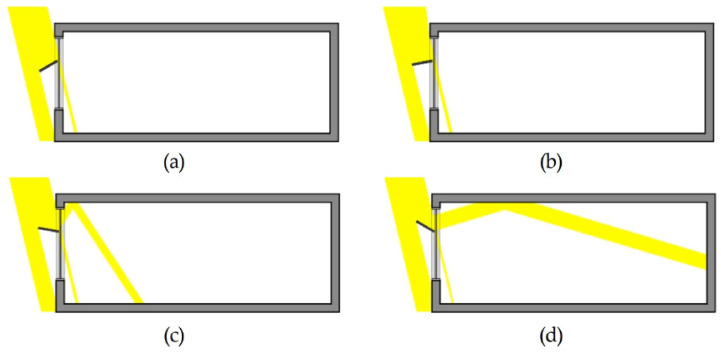
Diagram of natural light inflow into a room per light shelf angle in summer when the light shelf angle was (**a**) −30°, (**b**) −10°, (**c**) 10°, and (**d**) 30°.

**Figure 12 ijerph-18-02574-f012:**
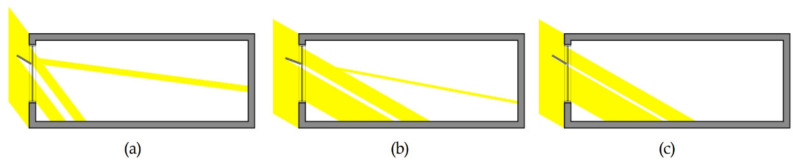
Diagram of natural light inflow into a room when the light shelf angle was (**a**) 30° in mid-season, (**b**) 20° in winter, and (**c**) 30° in winter.

**Figure 13 ijerph-18-02574-f013:**
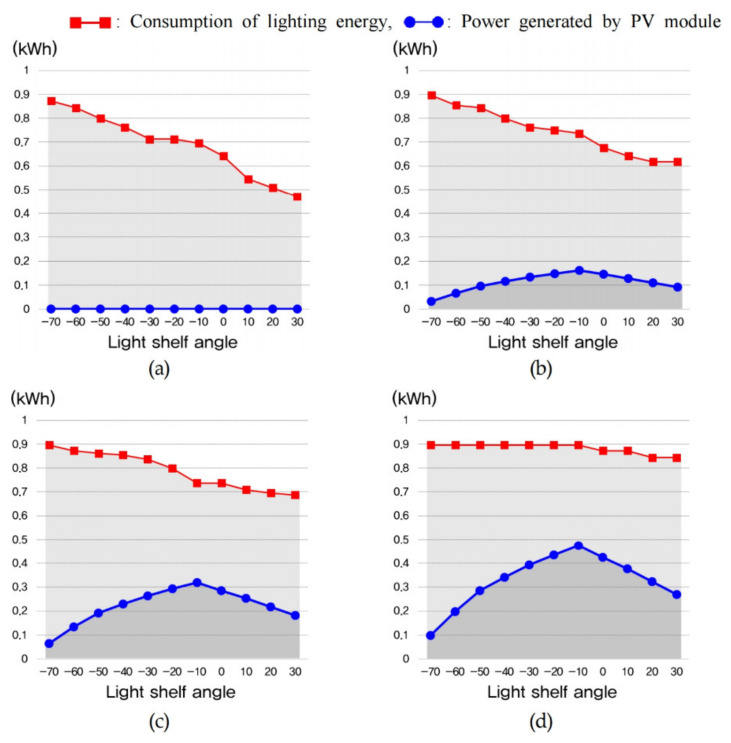
Consumption of lighting energy and power generated by the PV module to maintain optimal indoor illuminance during summer for (**a**) Case 2, (**b**) Case 3, (**c**) Case 4, and (**d**) Case 5.

**Figure 14 ijerph-18-02574-f014:**
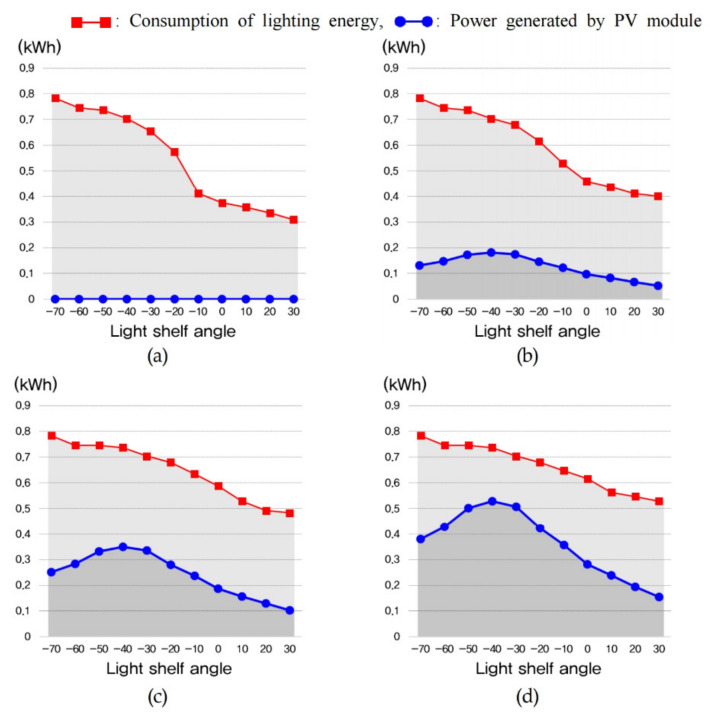
Consumption of lighting energy and power generated by the PV module to maintain optimal indoor illuminance during mid-season for (**a**) Case 2, (**b**) Case 3, (**c**) Case 4, and (**d**) Case 5.

**Figure 15 ijerph-18-02574-f015:**
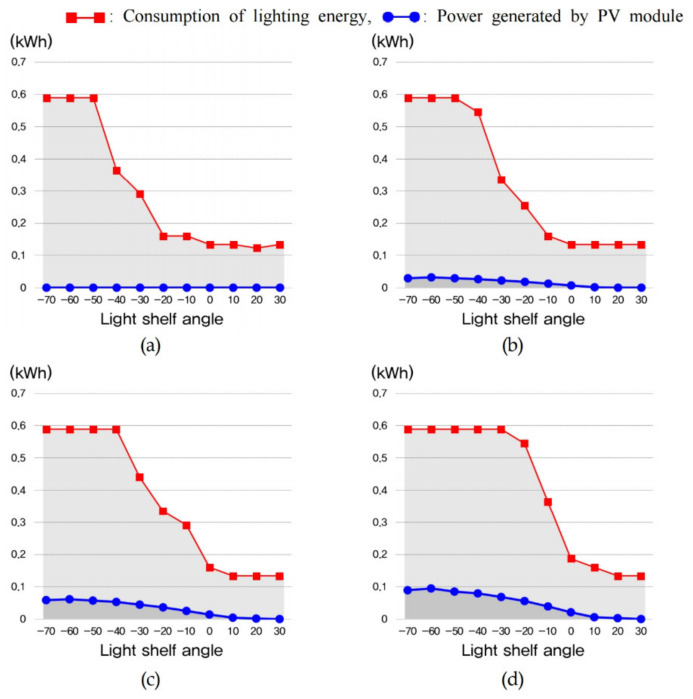
Consumption of lighting energy and power generated by the PV module to maintain optimal indoor illuminance during winter for (**a**) Case 2, (**b**) Case 3, (**c**) Case 4, and (**d**) Case 5.

**Figure 16 ijerph-18-02574-f016:**
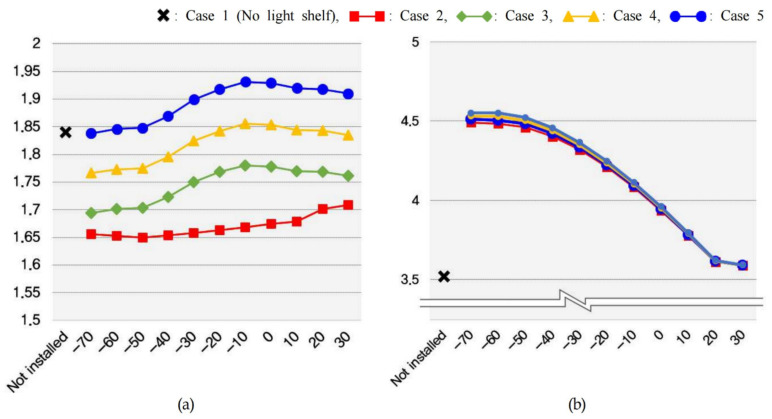
Consumption of (**a**) cooling energy during summer and (**b**) heating energy during winter to maintain the optimal indoor temperature.

**Figure 17 ijerph-18-02574-f017:**
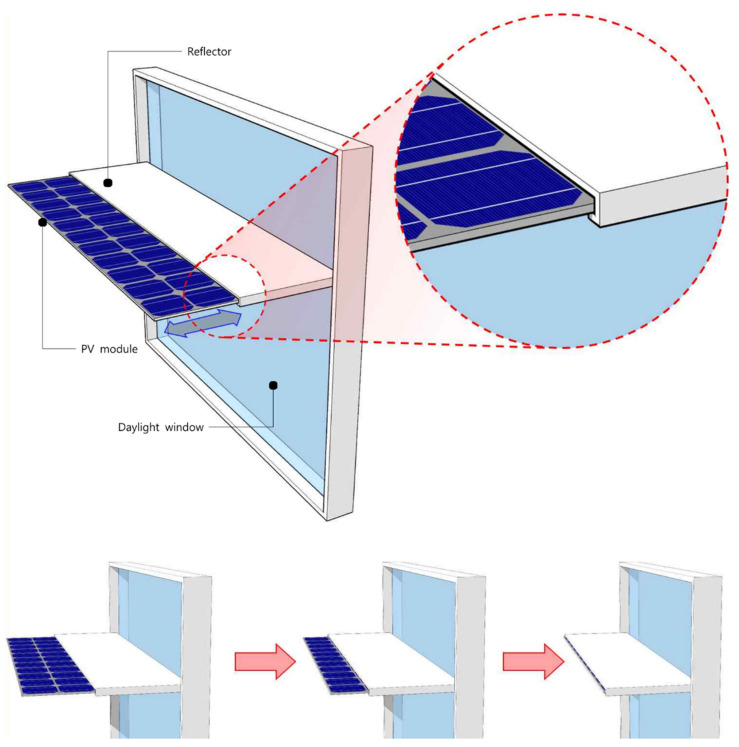
Proposed light shelf incorporating a PV module to save building energy.

**Table 1 ijerph-18-02574-t001:** Summary of prior light shelf research. PV, photovoltaic.

Author (Year)	Factors Considered to Improve Daylighting Performance		Primary Results	Consideration of Thermal Environment during Performance Evaluation
	Light Shelf Variable	Applied Technology		
Claros and Soler (2002) [16]	Reflectance	-	Indoor illumination distribution	Not considered
Hwang et al. (2014) [17]	-	PV module (attached to the front of the light shelf reflector)	Electricity generation	Not considered
Lin and Heng (2016) [18]	Height, reflector shape	-	Indoor illumination distribution	Not considered
Meresi (2016) [19]	Width, angle, type	-	Indoor illumination distribution	Not considered
Lee et al. (2016) [13]	Reflector shape	-	Indoor illumination distribution, lighting energy consumption	Not considered
Lee et al. (2016) [7]	Angle	Location awareness technology	Indoor illumination distribution, lighting energy consumption	Not considered
Berardi and Anaraki (2016) [20]	Light shelf installation	Window installation location and area	Indoor illumination distribution	Not considered
Lee et al. (2017) [14]	Width, angle	Perforated light shelf reflector	Indoor illumination distribution, lighting energy consumption	Not considered
Warrier and Raphael (2017) [21]	Reflectance (material)	-	Indoor illumination distribution, glare	Not considered
Lee et al. (2018) [15]	Angle	Awning + light shelf	Indoor illumination distribution, lighting energy consumption	Not considered
Lee et al. (2018) [22]	Reflectance, angle	Diffusion sheet	Indoor illumination distribution, glare, lighting energy consumption	Not considered
Kim et al. (2018) [8]	Angle	User awareness technology	Indoor illumination distribution, lighting energy consumption	Not considered
Lee (2019) [12]	Angle	PV module (adjusting PV attachment area)	Indoor illumination distribution, electricity generation, lighting energy consumption	Not considered
Mesloub and Ghosh (2020) [23]	Width, type, reflector type	PV module	Indoor illumination distribution, electricity generation	Not considered
Lee and Seo (2020) [5]	Angle	Prism sheet	Indoor illumination distribution, lighting energy consumption	Not considered
Lee (2020) [6]	Angle, reflectance	Changing reflectance through rolling technology	Indoor illumination distribution, glare, lighting energy consumption	Not considered

**Table 2 ijerph-18-02574-t002:** Optimal indoor illumination standards: general visual work. USA, United States of America.

Illumination Standards (Country)	Task Grade	Scope (lx)		
		Minimum Allowed Illumination	Standard Allowed Illumination	Maximum Allowed Illumination
IES (USA) [27]	General	500	750	1000
JIS Z 9110 (Japan) [28]		300	500	600
KS A 3011 (Republic of Korea) [29]		300	400	600

**Table 3 ijerph-18-02574-t003:** Illumination standards for office space (Europe and China). CAD, computer-aided design.

Illumination Standards	Specific Purpose	Average Illuminance (lx)
EN 12464-1 (Europe) [30]	Writing, typing, reading, data processing	500
	CAD workstations	500
	Conference and meeting rooms	500
GB 50034-2013 (China) [31]	Graphic design	500
	High-quality office space	500
	Sales office	500

**Table 4 ijerph-18-02574-t004:** Optimal indoor temperature standards.

Standards	Summer (°C)	Winter (°C)
ANSI/ASHRAE Standard 55-2013 (USA) [32]	23.0–26.0	20.0–23.5
ISO 10211: 2007 (Europe) [33]	23.0–26.0	20.0–24.0

**Table 5 ijerph-18-02574-t005:** Setting of cases for performance evaluation.

Case	Light Shelf Installation	Light Shelf Variables				Ratio of PV Modules Attached to the Light Shelf Reflector (Number of PV Cells)
		Width	Height	Angle	Reflectance	
1	×	-	-	-	-	-
2	O	0.52 m	Installed 1.8 m from the floor	10° increments, from −70° to 30°	85% (Specular reflection film)	0% (0)
3						33.3% (12)
4						66.9% (24)
5						100% (36)

**Table 6 ijerph-18-02574-t006:** Standard and efficiency grade of solar cell.

Type	Size (mm)	Power	Efficiency	Grade
Poly cell	156 × 156	4.43 W	18.2%	A

**Table 7 ijerph-18-02574-t007:** Test bed overview. LED, light-emitting diode; COP, coefficient of performance.

Room Size, Wall Material, and Reflexibility	
Size	4.9 m (W) × 6.6 m (D) × 2.5 m (H)
Wall material	Insulation panel (Thk 100 mm)
Reflexibility	Ceiling 86%, wall 46%, floor 25%
Window size and material	
Size	1.9 m (W) × 1.7 m (H)
Type	Double glazed 12 mm (3 CL + 6 A + 3 CL)
Thermal transmittance	2.83 W/m^2^·K (Summer), 2.69 W/m^2^·K (Winter)
Transmissivity	80%
Lighting	
Type	Eight-level dimming (LED type), four units
Dimensions (mm)	600 × 600
Dimming range	10%–100%
Energy consumption for phased light dimming	Dimming levels 1, 2, 3, 4, 5, 6, 7, and 8 are 12.3 kWh, 18.3 kWh, 22.0 kWh, 27.7 kWh, 34.0 kWh, 38.5 kWh, 42.6 kWh, and 50.8 kWh, respectively.
Air conditioner	
Model	AP-SM302 (EHP: Electric Heat Pump)
Capacity	Heating: 13,200 W; Cooling: 11,000 W
Energy consumption	Heating: 3.90 kW; Cooling: 3.90 kW
COP	Heating: 3.38; Cooling: 2.82
Illuminance sensor	
Sensing element	Silicon photo sensor, with filter
Detection range	0–200,000 lx
Precision	±3%
Temperature sensor	
Sensing element	NTC (Negative temperature coefficient thermistor) 10 kΩ: AN Type
Detection range	−40–90 °C
Precision	±0.3 °C
Artificial solar Light Radiation Apparatus	
Precision of solar light radiation	Grade-A (According to ASTM E927-85)
Range of illumination	0–80,000 lx
Directions	South aspect
Energy monitoring system	
Model	SPM-141
Measurement capacity	Single phase (220 V, 1–50 A)
Measurement items	Power/voltage/current, real-time, and accumulated amount
Error rate	Within 2.0%

**Table 8 ijerph-18-02574-t008:** External environment settings according to performance evaluation time.

Season		Time				
		10:00–11:00	11:00–12:00	12:00–13:00	13:00–14:00	14:00–15:00
Summer	Illuminance	70,000 lx	80,000 lx			70,000 lx
	Solar irradiation	429 W/m^2^	503 W/m^2^			429 W/m^2^
	Solar altitude	76.5°				
	Temperature	35 °C				
Middle Season	Illuminance	50,000 lx		60,000 lx		50,000 lx
	Solar irradiation	375 W/m^2^		359 W/m^2^		376 W/m^2^
	Solar altitude	52.5°				
	Temperature	21.2 °C				
Winter	Illuminance	20,000 lx		30,000 lx		20,000 lx
	Solar irradiation	283 W/m^2^		340 W/m^2^		283 W/m^2^
	Solar altitude	29.5°				
	Temperature	−11.3 °C				

**Table 9 ijerph-18-02574-t009:** Specifications and image of voltage and current measuring equipment. DC, direct current.

Equipment Name	Measurement Item (Measurement Capacity)	Error Rate
MULLER 3201	DC voltage (~6.000–1000 V), DC current (~6.000–10.00 A)	±0.5% + 3

**Table 10 ijerph-18-02574-t010:** Performance evaluation results by case: illuminance distribution and uniformity ratio. Min., minimum; Ave., average.

Case	Light Shelf Angle	Summer (External Illuminance: 80,000 lx)			Middle season (External Illuminance: 60,000 lx)			Winter (External Illuminance: 30,000 lx)		
		Illuminance Sensor (lx)		Uniformity Ratio	Illuminance Sensor (lx)		Uniformity Ratio	Illuminance Sensor (lx)		Uniformity Ratio
		Min.	Ave.		Min.	Ave.		Min.	Ave.	
1	Not installed	44.1	432.7	0.102	136.9	517.4	0.264	289.3	5552.1	0.052
2	−70	38.8	340.0	0.114	64.5	351.9	0.183	111.7	2678.9	0.042
	−60	40.4	343.5	0.118	66.9	353.4	0.189	124.8	2947.3	0.042
	−50	46.6	351.2	0.133	70.2	355.3	0.198	145.5	3408.1	0.043
	−40	54.4	359.8	0.151	76.3	358.1	0.213	193.3	4032.1	0.048
	−30	61.0	365.9	0.167	99.5	362.8	0.274	230.5	4678.3	0.049
	−20	68.2	376.5	0.181	123.8	368.3	0.336	245.1	4871.4	0.050
	−10	76.1	391.0	0.195	132.3	371.2	0.356	259.1	4987.4	0.052
	0	81.9	400.5	0.204	139.2	382.2	0.364	270.0	5089.2	0.053
	10	85.3	410.2	0.208	146.3	391.2	0.374	284.3	5187.6	0.055
	20	92.2	418.5	0.220	156.2	401.2	0.389	287.1	5452.0	0.053
	30	108.6	425.3	0.255	165.7	513.4	0.323	268.3	5181.9	0.052
3	−70	36.8	326.4	0.113	62.4	337.8	0.185	108.0	2639.0	0.041
	−60	37.6	329.8	0.114	64.7	339.3	0.191	120.6	2844.1	0.042
	−50	41.3	337.2	0.122	67.9	341.1	0.199	139.2	3271.8	0.043
	−40	47.1	345.4	0.136	73.8	343.8	0.215	167.6	3870.8	0.043
	−30	52.9	351.3	0.151	86.2	348.3	0.248	199.8	4491.1	0.044
	−20	59.1	361.4	0.164	107.3	353.6	0.303	212.4	4676.6	0.045
	−10	66.0	375.4	0.176	114.7	356.4	0.322	224.6	4787.9	0.047
	0	71.0	384.5	0.185	120.6	366.9	0.329	234.0	4885.6	0.048
	10	73.9	393.8	0.188	126.8	375.6	0.338	246.4	4928.3	0.050
	20	79.9	401.8	0.199	135.4	385.2	0.351	261.4	5288.5	0.049
	30	91.5	406.5	0.225	143.6	492.9	0.291	246.0	4974.1	0.049
4	−70	35.3	319.6	0.110	60.4	330.8	0.183	104.7	2531.3	0.041
	−60	35.6	322.9	0.110	62.7	332.2	0.189	116.9	2770.5	0.042
	−50	38.0	330.1	0.115	65.8	334.0	0.197	134.9	3203.6	0.042
	−40	41.0	338.2	0.121	71.5	336.6	0.212	163.1	3749.8	0.043
	−30	46.0	343.9	0.134	75.0	341.0	0.220	194.4	4350.8	0.045
	−20	49.1	353.9	0.139	89.1	346.2	0.257	205.3	4481.7	0.046
	−10	54.8	367.5	0.149	95.3	348.9	0.273	217.1	4688.1	0.046
	0	60.1	376.5	0.160	102.1	359.3	0.284	226.2	4783.8	0.047
	10	64.1	385.6	0.166	109.9	367.7	0.299	234.0	4800.2	0.049
	20	69.3	393.4	0.176	117.3	377.1	0.311	252.7	5124.9	0.049
	30	79.3	395.1	0.201	124.5	482.6	0.258	233.7	4871.0	0.048
5	−70	34.8	316.2	0.110	59.3	327.3	0.181	104.1	2517.9	0.041
	−60	35.2	320.3	0.110	63.0	335.7	0.188	112.3	2741.0	0.041
	−50	37.8	333.6	0.113	64.2	337.5	0.190	127.4	3067.3	0.042
	−40	39.8	341.8	0.116	67.5	340.2	0.198	153.8	3628.9	0.042
	−30	42.1	347.6	0.121	68.7	344.7	0.199	178.9	4163.7	0.043
	−20	47.1	357.7	0.132	85.4	349.9	0.244	188.9	4286.9	0.044
	−10	50.4	371.5	0.136	87.6	352.6	0.249	199.7	4488.7	0.044
	0	55.3	368.5	0.150	93.9	351.6	0.267	208.1	4580.2	0.045
	10	58.9	377.4	0.156	101.1	359.9	0.281	215.3	4722.9	0.046
	20	63.7	385.0	0.165	107.9	369.1	0.292	232.5	5015.9	0.046
	30	73.0	386.7	0.189	114.5	472.3	0.242	215.0	4767.3	0.045

**Table 11 ijerph-18-02574-t011:** Consumption of lighting energy to maintain optimal indoor illuminance with no light shelf installed.

Season	External Illuminance (lx)	Lighting Dimming Control: Light Number (Dimming Level)	Consumption of Lighting Energy (kWh)
Summer	80,000	1(8) + 3(8) + 2(5)	0.712
	70,000	1(8) + 3(8) + 2(8)	
Middle Season	60,000	1(8) + 3(4)	0.411
	50,000	1(8) + 3(5)	
Winter	30,000	1(2)	0.123
	20,000	1(5)	

**Table 12 ijerph-18-02574-t012:** Optimal specifications considering lighting energy savings and PV generation during summer.

Case	Optimal Light Shelf Angle	Lighting Energy Consumption (kWh)	Power Generated by PV Module (kWh)	Total Energy Consumption (kWh)	Energy-Saving Ratio Compared to No Light Shelf
		*A*	*B*	*∑ A − B*	
2	30	0.471	0	0.471	33.8% reduction
3	20	0.618	0.110	0.508	28.7% reduction
4	−10	0.737	0.318	0.419	41.2% reduction
5	−10	0.896	0.473	0.423	40.6% reduction

**Table 13 ijerph-18-02574-t013:** Optimal specifications considering lighting energy savings and PV generation during mid-season.

Case	Optimal Light Shelf Angle	Lighting Energy Consumption (kWh)	Power Generated by PV Module (kWh)	Total Energy Consumption (kWh)	Energy-Saving Ratio Compared to No Light Shelf
		*A*	*B*	*∑ A − B*	
2	20	0.335	0	0.335	18.5% reduction
3	20	0.411	0.067	0.344	16.3% reduction
4	20	0.492	0.129	0.363	11.7% reduction
5	−30	0.704	0.508	0.196	52.3% reduction

**Table 14 ijerph-18-02574-t014:** Optimal specifications considering lighting energy savings and PV generation during winter.

Case	Optimal Light Shelf Angle	Lighting Energy Consumption (kWh)	Power Generated by PV Module (kWh)	Total Energy Consumption (kWh)	Energy-Saving Ratio Compared to No Light Shelf
		*A*	*B*	*∑ A − B*	
2	10	0.134	0	0.134	8.9% increase
3	10	0.134	0.007	0.127	3.3% increase
4	10	0.134	0.004	0.130	5.7% increase
5	10	0.160	0.006	0.154	25.2% increase

**Table 15 ijerph-18-02574-t015:** Performance evaluation results by case: energy consumption and PV power generation to maintain the optimal indoor environment.

Case	Light Shelf Angle	Summer				Winter			
		L.C. (kWh)	P.G. (kWh)	C.C. (kWh)	T.C. (kWh)	L.C. (kWh)	P.G. (kWh)	C.C. (kWh)	T.C. (kWh)
		*A*	*B*	*C*	*∑ A + B − C*	*A*	*B*	*C*	*∑ A + B − C*
1	Not installed	0.712	0	1.840	2.552	0.123	0	3.521	3.644
2	−70	0.872	0	1.656	2.528	0.589	0	4.492	5.081
	−60	0.843	0	1.653	2.496	0.589	0	4.485	5.074
	−50	0.799	0	1.650	2.449	0.589	0	4.461	5.050
	−40	0.762	0	1.654	2.416	0.364	0	4.404	4.768
	−30	0.712	0	1.658	2.370	0.291	0	4.320	4.611
	−20	0.712	0	1.663	2.375	0.160	0	4.211	4.371
	−10	0.695	0	1.668	2.363	0.160	0	4.084	4.244
	0	0.642	0	1.674	2.316	0.134	0	3.936	4.070
	10	0.545	0	1.678	2.223	0.134	0	3.778	3.912
	20	0.508	0	1.701	2.209	0.123	0	3.612	3.735
	30	0.471	0	1.709	2.180	0.134	-	3.590	3.724
3	−70	0.896	0.033	1.694	2.557	0.589	0.030	4.512	5.071
	−60	0.854	0.067	1.701	2.488	0.589	0.032	4.507	5.064
	−50	0.843	0.097	1.703	2.449	0.589	0.029	4.482	5.042
	−40	0.799	0.116	1.723	2.406	0.545	0.027	4.422	4.940
	−30	0.750	0.133	1.750	2.367	0.335	0.023	4.335	4.647
	−20	0.712	0.148	1.768	2.332	0.254	0.019	4.223	4.458
	−10	0.695	0.161	1.780	2.314	0.160	0.013	4.094	4.241
	0	0.668	0.145	1.778	2.301	0.134	0.007	3.944	4.071
	10	0.642	0.128	1.770	2.284	0.134	0.002	3.784	3.916
	20	0.618	0.110	1.768	2.276	0.134	0.001	3.615	3.748
	30	0.618	0.092	1.761	2.287	0.134	-	3.591	3.725
4	−70	0.896	0.065	1.766	2.597	0.589	0.059	4.532	5.062
	−60	0.872	0.133	1.773	2.512	0.589	0.062	4.529	5.056
	−50	0.843	0.192	1.775	2.426	0.589	0.057	4.502	5.034
	−40	0.806	0.229	1.795	2.372	0.589	0.053	4.440	4.976
	−30	0.799	0.264	1.824	2.359	0.441	0.045	4.350	4.746
	−20	0.762	0.293	1.842	2.311	0.335	0.037	4.235	4.533
	−10	0.695	0.318	1.855	2.232	0.291	0.025	4.103	4.369
	0	0.695	0.286	1.853	2.262	0.160	0.014	3.952	4.098
	10	0.668	0.254	1.844	2.258	0.134	0.004	3.790	3.920
	20	0.668	0.217	1.843	2.294	0.134	0.002	3.618	3.750
	30	0.642	0.182	1.835	2.295	0.134	-	3.592	3.726
5	−70	0.896	0.098	1.838	2.636	0.589	0.089	4.552	5.052
	−60	0.896	0.197	1.846	2.545	0.589	0.095	4.551	5.045
	−50	0.896	0.285	1.848	2.459	0.589	0.086	4.522	5.025
	−40	0.896	0.340	1.869	2.425	0.589	0.080	4.457	4.966
	−30	0.896	0.393	1.899	2.402	0.589	0.068	4.365	4.886
	−20	0.896	0.435	1.918	2.379	0.545	0.056	4.247	4.736
	−10	0.896	0.473	1.931	2.354	0.364	0.039	4.113	4.438
	0	0.896	0.425	1.929	2.400	0.187	0.021	3.960	4.126
	10	0.872	0.377	1.920	2.415	0.160	0.006	3.795	3.949
	20	0.843	0.322	1.918	2.439	0.134	0.003	3.621	3.752
	30	0.843	0.270	1.910	2.483	0.134	-	3.593	3.727

L.C.: lighting energy consumption, P.G.: PV module generation, C.C.: cooling energy consumption, T.C.: total energy consumption.

**Table 16 ijerph-18-02574-t016:** Performance evaluation results by case: energy consumption and PV power generation to maintain an optimal indoor environment.

Case	Season	PV Module Light Shelf without Considering Heating and Cooling Energy			PV Module Light Shelf Considering Heating and Cooling Energy		
		Optimal Angle	E.C. (kWh)	Total Energy Consumption (kWh)	Optimal Angle	E.C. (kWh)	Total Energy Consumption (kWh)
2	Summer	30	0.471	0.940	30	2.180	6.427
	Middle season	20	0.335		20	0.335	
	Winter	10	0.134		10	3.912	
3	Summer	20	0.508	0.978	20	2.276	6.536
	Middle season	20	0.344		20	0.344	
	Winter	0	0.127		10	3.916	
4	Summer	−10	0.377	0.870	−10	2.232	6.515
	Middle season	20	0.363		20	0.363	
	Winter	10	0.130		10	3.920	
5	Summer	−10	0.423	0.773	−10	2.354	6.499
	Middle season	−30	0.196		−30	0.196	
	Winter	10	0.154		10	3.949	

E.C. (energy consumption) = (lighting energy) − (PV module power generation).

## Data Availability

No additional data available.

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
