# Peer review of "A Study of Optimal Specifications for Light Shelves with Photovoltaic Modules to Improve Indoor Comfort and Save Building Energy"

_ijerph, 2021, doi:10.3390/ijerph18052574_

Round 1

Reviewer 1 Report

The article, presenting results from "a study of optimal specifications for light shelves with photovoltaic modules to improve indoor comfort and save building energy", is an interesting contribution within the currently growing area of dynamic design towards the Green Shift in Built Environment. However, relatively minor fixes, listed below, should be considered by the authors:

1) Professional language/the use of expressions could be more caring and accurate as well as more cosequently used, for example in a definition of PV "concentration of light" or "the heat generated by PV..." could be exchanged by "absorbing and converting sunlight into electricity/electrical energy", or so.

2) Some important definitions/expressions used frequently, such as for example  "the uniformity ratio", "PV" (please, note PV or BIPV?), could be presented and explained more clearly. Most preferably, formal definitions could be quoted with respective references.

3) Some limitations for the results relevance, for example associated with climate zones, should be addressed.

4) Legal conditions: why only US standards are addressed (36-37), not other standards, especially relevant for Korea?

5) Standards: in the standard addressed by the authors (thermal comfort), the optimal indoor temperature/summer is 23-26 degrees. Why the authors founded the temperature of 26 degrees as optimal/summer in this study? It can be considered too high, it should be the maximal allowed temperature for comfort.

6) Chapter 2 and Chapter 3: although the results are presented in detail, they could be presented more structured and clearly in terms of the correspondence between descriptions and illustrations (tables and figures) using more accurate drawings or completing existing.

6.1) Chapter 2.5: "Performance evaluation method". This section could be started with an easy diagram showing/explaining the method step-by-step, then completed with the following text (existing).

7) Graphics: Some tables are nice presented (2,3,4) while some could be improved (2, 5, 7, 10, ...). Some drawings could be bigger/completed. Figures 1 and 2 could be presented together. Figure 3 is missing descriptons (a,b,c). Figure 5: "...Case 2, light shelf angle 0..." could not be repeted. (Additionally, other cases could be illustrated with sections). Figure 8: "...plane, section...", not "...section, plane..." or "plane"/"section" placed directly close to the respective drawings. And more ...

8) Proposal: The "proposed light shelf incorporating a PV module to save building energy" (Figure 17) is an interesting proposal contributing to the idea of architectural design flexibility. However it could be more exciting presented in terms of its flexibility. It is up to the authors creativity, and I believe it will be presented showing this proposal full potential in the final version. The aesthetic aspect should be paid more attention.

I read the article with pleasure, and I hope my comments will contribute to improving it. I can't wait the next and published version.

Author Response

According to the reviewers’ comments, the manuscript underwent minor revisions. All the changes are marked in the resubmitted manuscript. We elaborate on how we did this, and how we respond to the reviewer’s comments in a separate file attached.

Reviewer 2 Report

In this work, a study of photovoltaic modules applied to light shelves has been carried out in terms of indoor comfort and building energy saving. In my opinion, the manuscript is a good contribution to the huge field of energy performance in buildings.

I only suggest a better organization of the two sections of the manuscript: 3.2 Discussion and 4. Conclusion. They have basically the same amount of information. For example, in the conclusion, the authors could propose further studies aiming at evaluating the economic impact of the different analyzed cases (angles, numbers of PV modules, etc.).

Author Response

(The authors gave the same response as above.)
